# Phospholipid-Based Microemulsions for Cutaneous Imiquimod Delivery

**DOI:** 10.3390/ph15050515

**Published:** 2022-04-22

**Authors:** Eleni Panoutsopoulou, Jarmila Zbytovská, Kateřina Vávrová, Georgios Paraskevopoulos

**Affiliations:** 1Skin Barrier Research Group, Faculty of Pharmacy in Hradec Králové, Charles University, Akademika Heyrovského 1203, 50005 Hradec Králové, Czech Republic; panoutse@faf.cuni.cz (E.P.); katerina.vavrova@faf.cuni.cz (K.V.); 2Department of Pharmaceutical Technology, Faculty of Pharmacy in Hradec Králové, Charles University, Akademika Heyrovského 1203, 50005 Hradec Králové, Czech Republic; jarmila.zbytovska@vscht.cz; 3Faculty of Chemical Technology, University of Chemistry and Technology, Technická 5, 16628 Prague, Czech Republic

**Keywords:** microemulsions, imiquimod, dermal delivery, oleic acid, transepidermal water loss, infrared spectroscopy

## Abstract

Imiquimod (IMQ) is a potent immune response modifier with antiviral and antitumor properties. IMQ’s low aqueous solubility and unsatisfactory cutaneous permeability limit its formulation into effective dosage forms. This work aimed to develop IMQ-loaded microemulsions (MEs) based on phospholipids and oleic acid to improve IMQ penetration into the epidermis. A pseudo-ternary phase diagram was constructed, and the microstructure of the formulations was examined by measuring the conductivity values. Selected MEs were characterized and studied for their ability to deliver IMQ into and through ex vivo human skin. ME_1_ with 1% IMQ (bicontinuous ME with Bingham rheology) delivered similar IMQ quantities to the human epidermis ex vivo as the commercial product while having a 5-fold lower IMQ dose. IMQ was not detected in the acceptor phase after the permeation experiment, suggesting a lower systemic absorption risk than the established product. Infrared spectroscopy of the *stratum corneum* revealed less ordered and less tightly packed lipids after ME_1_ application. The ME_1_-induced barrier disruption recovered within less than 5 h after the formulation removal, as detected by transepidermal water loss measurements. In conclusion, our findings demonstrate that phospholipid and oleic acid-based MEs could become a promising alternative for topical IMQ administration.

## 1. Introduction

Imiquimod (IMQ) is an immune response modifier clinically used to treat genital and perianal warts, actinic keratosis, and superficial basal cell carcinoma [1]; this compound is also experimentally used to induce psoriasis-like skin inflammation in mouse models [2,3,4], and as an adjuvant to weakly immunogenic vaccines [5,6,7]. Despite being a small molecule (molar mass 240 g/mol), IMQ has low aqueous solubility and unsatisfactory cutaneous permeability, limiting its formulation into effective dosage forms [8]. The marketed formulations usually contain 5% IMQ; however, reports indicate that only less than 2% of the applied dose reaches human skin [1,9]; thus, the unabsorbed formulation contains a potentially hazardous IMQ dose, both to the patient and the environment. In addition, part of the IMQ dose can reach systemic circulation and cause back pain, influenza-like symptoms, fatigue, headache, and other side effects [4,10,11]. Several carriers for improving the safety and efficacy of IMQ have been developed, such as complexes, micelles, liposomes, and nanostructured vehicles [7,8,12]. Among various skin delivery systems, nano- and microemulsion (ME) systems have gained recent popularity for their efficacies in delivering different active substances [1,13,14,15,16,17].

In this study, we probed the potential of ΜΕs for improving the IMQ delivery into human skin. MEs are isotropic, thermodynamically stable, transparent mixtures of water, oil, and surfactants [18]; they are easy to prepare, have a long shelf life, and considerably enhance drug penetration [1,19,20,21,22,23]. The increased drug penetration provided by MEs can be explained by one or more of the following effects: their small particle size (up to 100 nm), increased skin hydration, high drug loading capacity, and the action of the individual constituents [24,25].

The choice of the ME oil phase may impact the drug solubilization but also skin permeability [24]. IMQ is well soluble in oleic acid (OA), a common ME constituent for topical drug delivery. OA acts as a potent permeation enhancer, fluidizing the skin barrier lipids and forming more permeable domains [23,26]. In addition, OA, due to its amphiphilic structure, has been used for the formulation of microemulsions without the use of a classical surfactant [27]. Phospholipids (PL) such as lecithin have been included in ME technology as surfactants for their low toxicity and penetration-enhancing properties [28,29,30]. Lecithin is a good emulsifier that can change skin lipid fluidity [26,30]. In PL-based MEs, short-chain alcohols, e.g., ethanol, have been previously used as cosurfactants [21,30]. Ethanol reduces the rigidity of the lecithin film and decreases the interfacial tension in the formulation while also acting as a penetration enhancer [21,30,31].

This study aimed to develop a stable PL-based ME that would (1) contain less than 5% IMQ, (2) deliver IMQ into the human epidermis and dermis in amounts comparable to the commercially available 5% IMQ cream, and (3) not irreversibly compromise the skin barrier properties. As the previous studies on IMQ-loaded MEs used porcine skin and infinite formulation doses (200 mg/cm^2^) under occlusion, we decided to probe the performance of MEs on human skin with dose (25 µL/cm^2^, unoccluded) and application time according to the commercial formulation’s instructions (8 h on, 16 h off) to mimic the clinical application to a greater extent [1,13,14]. To obtain insight into the mechanisms of action of selected MEs, we probed their interactions with the human *stratum corneum* (SC) using infrared spectroscopy.

## 2. Results and Discussion

### 2.1. Pseudo-Ternary Phase Diagram

IMQ is a highly lipophilic drug with low solubility in aqueous solvents [32], making its formulation challenging [30]. We confirmed that IMQ solubility in water is very low (0.002 mg/mL). In contrast, we found that IMQ was highly soluble in OA (80 mg/mL), consistent with published reports [1]. Long-chain fatty acids likely solubilize IMQ due to their lipophilicity, potential ion pairing, and hydrogen bond formation with the drug [1,32]; thus, OA was chosen in this study as an oil phase with multiple actions due to its amphiphilic character, ME creation ability and IMQ solubility enhancement. To identify the ME region [33], a pseudo-ternary diagram was built, using water as the aqueous phase, OA as the oil phase, and PL/ethanol 1:1 *w*/*w* as the surfactant and co-surfactant mixture S_mix_ (Figure 1a)

MEs were characterized as clear, transparent formulations with low viscosity; the ME region is highlighted in blue in Figure 1a. MEs were formed when the water content was less than approximately 20% (regardless of the OA: S_mix_ ratio). Further addition of water resulted in a coarse, opaque mixture resembling an emulsion, likely due to PL self-assembly into bilayers (e.g., liposomes) in aqueous media [34]. Figure 1a also shows the composition of the selected MEs to which IMQ was loaded; for details, see Table 1. The MEs were selected for their ability to dissolve various IMQ amounts. HPLC confirmed that the added amount was completely incorporated in the formulations; thus, the encapsulation efficiencies were 100% (complete dissolution). ME_5.2_, ME_2.5_, and ME_3.75_ had similar drug concentrations as the clinically used formulations [4,35,36].

### 2.2. Conductivity Studies

Electrical conductivity is a crucial parameter to characterize the ME microstructure [37]. The gradual water addition and dilution of the system result in conductivity changes, which allow the ME type identification (Figure 1b). At low OA content (OA/S_mix_ ratios of 0.5:9.5–2:8), the MEs have a conductivity similar to S_mix_ (κ = 0.25 mS/cm), and this value decreases with a higher OA fraction. With water addition, the conductivity of these samples increases linearly, suggesting an increasing number of water channels enabling the motion of ions and the existence of a percolation phenomenon that may be attributed to the inverse microdroplet aggregation [37]; this behaviour indicates that water droplets fuse to form a continuous phase, and the aqueous and oil phases co-exist in a pool-like state, creating a bicontinuous microstructure. Such ME microstructure may be attributed to high ethanol content in the S_mix_, which reduces the interfacial rigidity [30,31], allowing the droplets to merge with the continuous phase and favouring the formation of bicontinuous ME with low water content [38].

The samples with high OA/S_mix_ ratios (6:4–9:1) have conductivity values close to zero irrespective of water addition across the whole ME range, indicating water in oil (*w*/*o*) MEs containing small, dispersed water droplets. The external oil phase prevents the migration of ions and leads to low electrical conductivity.

In the MEs with the 3:7–5:5 OA/S_mix_ ratios, the conductivities suggest *w*/*o* MEs at <5% water; however, higher water content increases conductivity, which then plateaus, and, in some cases, decreases; this phenomenon may be attributed to the percolation effect, which causes a sudden increase in conductivity; thus, the samples become bicontinuous at approximately 5–15% water. The conductivity decrease at around 15% water corresponds to the appearance of water-continuous microemulsion-type media and can be explained by the formation of *o*/*w* ME with decreased concentration of the dispersed oil droplets [39].

Concluding, formulations that were characterized as MEs had always less than approximately 20% of water content regardless of the OA: S_mix_ ratio. Furthermore, MEs were divided into *o*/*w*, bicontinuous or *w*/*o* according to their conductivity values. More specifically, MEs with low OA content and OA/S_mix_ ratios of 0.5:9.5–2:8 were characterized as *o*/*w* (Figure 1a—included in the grey region). MEs with medium OA/S_mix_ ratios ranging between 3:7–5:5 were characterized either as *o*/*w* (when water contact was around 15%, Figure 1a—included in the grey region), bicontinuous (when water content was 5–15%, Figure 1a—blue region) or *w*/*o* (when water content was <5%, Figure 1a—included in green region). Finally, MEs with high OA/S_mix_ ratios (6:4–9:1) were characterized as *w*/*o* (Figure 1a—included in the green region).

### 2.3. Microemulsion Characterization, Rheological Behaviour, and Stability Studies

Selected IMQ-loaded MEs were characterized for their size (particle diameter). Figure 2 shows the size characteristics of the MEs obtained from dynamic light scattering (DLS) as a function of OA/S_mix_ ratios. Even though the systems were identified as bicontinuous, we found a relatively narrow size distribution of the internal MEs structure. The characteristics interpreted as ME size increase as the formulation approaches the bicontinuous structure at lower oil content [40]. In our MEs, the size characteristics increased with the rising S_mix_ amount (especially in the case of bicontinuous ME_0.6_ and ME_1_). The higher ethanol amounts in these samples make the interfacial film more flexible and increase the single-phase area, allowing the aqueous droplets to enlarge [38].

No phase separation or aggregation was observed after centrifugation, and the preparations were stable for at least 10 weeks. The DLS characteristics of the selected MEs did not change for 10 weeks, suggesting the stability of the formulations.

Rheology studies showed that most of the studied MEs (ME_0.3_, ME_2.5_, ME_3.75_, and ME_5.2_) exhibited Newtonian flow. A linear correlation between shear stress and the shear rate, intersecting zero [30] (Figure 3), was observed, which is typical of MEs [28]. The MEs viscosities increased with increasing OA content and decreasing amount of S_mix_, except for ME_1_. Notably, ME_0.6_ and ME_1_ exhibited a Bingham rheology, where the shear stress must exceed a specific yield stress value before the fluid deforms and flows [41]; this behaviour may be related to the higher water content in those MEs (~20%) compared to formulations with the Newtonian flow. The Bingham behaviour has been previously described in MEs and was suggested to be related to an internal structure of flocculated particles in concentrated dispersed systems. The saturation of these two formulations with water and drug could generate high frictional forces between the moving droplets, eventually leading to the yield value [41].

### 2.4. IMQ Permeation through and into Human Skin

When applied topically, IMQ should ideally be accumulated in the viable epidermis (to activate Langerhans cells primarily located in the *stratum spinosum*) [1,42], with negligible systemic absorption. The ME effects on the IMQ delivery through and into different skin layers were evaluated using human skin, and the commercial formulation Aldara was used as a positive control. The formulations (25 µL/cm^2^) were applied to the skin for 8 h and then washed away to mimic the recommended application. The specific amount was chosen according to the FDA recommendations for Aldara. More specifically, 25 mg is the recommended Aldara amount to be used when the application area’s diameter is between 1 to 1.5 cm. The IMQ amount in the epidermis, dermis and acceptor compartment was determined 24 h after initial application (16 h after formulations were washed away) (Figure 4).

Importantly, none of the prepared MEs delivered IMQ into the acceptor phase in detectable concentrations, in contrast to Aldara (Figure 4a). Although in vitro permeation experiments cannot fully predict the in vivo systemic drug absorption, these results suggest that our IMQ-loaded MEs have a decreased risk of systemic side effects, previously reported for the commercially available products [4].

ME_5.2_, which contained a similar IMQ concentration as Aldara (~5%), delivered approximately twice more IMQ into the epidermis (0.78 μg IMQ per mg of tissue) and dermis than this clinical formulation (Figure 4a–c); thus, despite being lipophilic, substantial IMQ amounts can reach the viable epidermis and even dermis. To get insight into the IMQ partitioning between the SC and viable epidermis, we tape-stripped Aldara-treated skin and found approximately equal IMQ amounts in the SC and viable epidermis; this better performance of ME_5.2_ over Aldara may be attributed to the high OA concentration (61.6%) in ME_5.2_. OA is a potent permeation enhancer widely used for transdermal drug delivery [26,43,44]. OA can increase skin permeability by disordering the highly packed SC intercellular lipids or forming separate fluid pools in the barrier lipids (which provide a more permeable shortcut for drugs) [45,46,47]. In addition, ethanol can significantly enhance drugs’ cutaneous deposition when combined in vesicles with an additional penetration enhancer, such as OA [47,48,49].

MEs with Newtonian rheology (ME_0.3_, ME_2.5_, ME_3.75_, and ME_5.2_) were found to deliver IMQ proportionally to their IMQ concentrations; thus, the drug concentration gradient was likely driving their performance. On the other hand, MEs with Bingham rheology did not follow this trend and delivered more than expected given the formulations’ drug amount (particularly ME_1_; Figure 4a).

Notably, ME_1_ with 1% IMQ and Aldara (5% IMQ), delivered comparable IMQ amounts into the epidermis and dermis (ca 0.42 and 0.29 μg IMQ per mg of the epidermis, respectively). Considering the percentage of applied dose that reached the epidermis (Figure 4d), it was only 0.15% for Aldara and 1.01% for ME_1_. Although these percentages are far from ideal, ME_1_ would also be associated with a decreased risk over Aldara from the unabsorbed formulation (that may, for example, get accidentally ingested, transferred to a child, or contaminate the environment) [50].

### 2.5. Reversibility of MEs Effects on Transepidermal Water Loss (TEWL)

Formulations should increase drug permeation without irreversibly compromising the skin’s barrier properties; thus, we probed the ME effects on TEWL (Figure 5), a widely used estimate of the skin’s water permeability barrier [51]. Figure 5a shows that all MEs and Aldara (8-h application) significantly increased TEWL and maximum TEWL was induced by ME_1_ (a formulation containing approximately 12% OA, 34% PL, and 34% ethanol). ME_2.5_, ME_3.75_, and ME_5.2_ (with more OA and less PL/ethanol mixture) were less effective in increasing TEWL than ME_1_, suggesting that PL and/or ethanol are more TEWL disturbing components than OA. For further experiments, we selected ME_1_ (which delivered the same as Aldara with less IMQ dose), ME_5.2_ (having a similar dose as Aldara), and Aldara (as a positive control). We followed the TEWL values induced by these formulations in time and found that the water barrier disruption is temporary and rapidly returns to baseline values (within 5 h after the formulation removal from the skin; Figure 5b). Considering that the environmental conditions could potentially affect TEWL values, untreated skin samples were used as a negative control, and temperature and humidity levels remained stable throughout the measurements [52]. We have previously shown that such a setup can distinguish reversible and irreversible effects [53].

### 2.6. Interactions of Selected Formulations with the Human SC Studied by FTIR Spectroscopy

To obtain an insight into the mechanism(s) by which the selected MEs decreased the skin barrier properties, isolated human SC treated with those formulations was studied by FTIR spectroscopy. FTIR spectroscopy is a sensitive and non-perturbing tool to investigate the interactions of compounds with SC lipids or proteins [54]. First, we searched for signs of lipid extraction as the MEs contained ethanol. Figure 6a shows that none of the formulations decreased the methylene symmetric stretching band intensity (originating mainly from lipid chains) relative to the amide I band intensity (mostly from peptide bonds in SC proteins), indicating that the formulations do not extract SC lipids. The formulations did not induce any significant change in protein conformation as deduced from the amide bands (not shown). The lack of lipid extraction and protein changes is consistent with the observed rapid reversibility of TEWL.

In contrast, the relative intensity of lipid bands significantly increased in ME_1_-treated SC over untreated control, indicating penetration of ME_1_ lipid constituents into SC (Figure 6a); this assumption is further supported by the bands that appear in the ME_1_-treated SC (and are absent or weak in untreated SC): 3009 cm^−1^ (=CH stretching, attributable either to OA or unsaturated PL), 1739 cm^−1^ (PL’s ester carbonyl stretching), 1084 (PL’s PO_2_ symmetric stretching, putatively combined with ethanol C-O stretching), 1062 cm^−1^ (PL’s PO_2_ stretching, putatively combined with ethanol C-O stretching), 970 cm^−1^ (N+(CH_3_)_3_ from choline in PL), 822 cm^−1^ (asymmetric PO stretching). These bands indicate that PL molecules from ME_1_ penetrate the SC (although we cannot exclude some OA and ethanol penetration as their characteristic bands overlap with PL).

The penetration of ME_1_ components into the SC was accompanied by decreased lipid chain order and less tight chain packing (Figure 6a–c). Figure 6b shows a significant shift of methylene symmetric stretching vibration from 2848.8 cm^−1^ in control (untreated) SC samples to 2851.0 cm^−1^ in ME_1_-treated SC (along with a peak broadening from 7.7 ± 0.5 cm^−1^ to 13.1 ± 3.1 cm^−1^, respectively), indicating an increased proportion of *gauche* conformers in the lipid chains [55]. Such behaviour could be interpreted either as SC lipid fluidization induced by ME_1_ components or creating fluid domains (probably PL-rich) within the SC lipids—without deuterated ME components, we cannot distinguish between those mechanisms; nevertheless, such higher hydrocarbon chain mobility in the ME_1_-treated SC would enhance drug diffusion [56], consistent with the highest IMQ bioavailability and TEWL change induced by ME_1_. There were no significant shifts of the methylene symmetric stretching vibration in ME_5.2_ or Aldara-treated samples.

In addition, human SC lipids are primarily arranged in a tight orthorhombic packing, which is vital for restricting water loss [57]; this chain packing can be detected as the correlation field splitting of the scissoring or rocking methylene bands. The rocking doublet at around 730 and 718 cm^−1^ is usually well resolved in the complex SC samples. Figure 6c shows the disappearance of the 730 cm^−1^ doublet component in the ME_1_ and Aldara-treated samples, indicating a disruption of the tight orthorhombic lipid packing [58]. The doublet persisted in most of the ME_5.2_-treated samples. These data are consistent with the formulations’ effects on TEWL.

## 3. Materials and Methods

### 3.1. Materials

Imiquimod (IMQ, purity > 98%, molar mass 240.31 g/mol) and oleic acid (OA, >99%) were purchased from TCI chemicals (Tokyo, Japan). Phospholipids (PL) from egg yolk (60% phosphatidylcholine, mostly 16:0/18:2, 15% phosphatidylethanolamine, mostly 18:0/18:2 and 18:0/20:4, minor fractions phosphatidylinositol, lysophosphatidylcholine, sphingomyelin, lysophosphatidylethanolamine), acetic acid (>99%), ethanol (>99%), acetonitrile (HPLC grade), and methanol (HPLC grade) were purchased from Sigma Aldrich (Schnelldorf, Germany). Aldara topical cream (5% IMQ) MEDA company, Sweden was obtained from Hospital Pharmacy, Hradec Králové, Czech Republic. Water was deionized, distilled, and filtered through a Milli-Q RG water purification system (Millipore, Burlington, MA, USA).

### 3.2. Skin

Human skin from Caucasian female individuals who underwent abdominal plastic surgery was used with the approval of the Ethics Committee of the Sanus Surgical Centre in Hradec Králové (No. 5/4/2018), according to the principles of the Declaration of Helsinki. Written informed consent has been obtained from all individuals. The subcutaneous fat tissue was carefully removed, and the remaining full-thickness skin was washed with saline, blotted dry, and stored at −20 °C.

### 3.3. Imiquimod HPLC Quantification Method

IMQ was determined by a published HPLC method [1] with a slight modification, using an Agilent 1200 series instrument (Agilent Technologies, Waldbronn, Germany) equipped with a G1379B degasser, G1310A isocratic pump, G1316A thermostatted column, G1329A autosampler, and G1321B fluorescence detector. IMQ was analysed on an HS Discovery C-18 150 mm × 4.6 mm column (5 μm particles with a 100 Å porosity) at 25 °C. The mobile phase consisted of methanol/acetonitrile/acetate buffer (100 mM, pH = 4) 185:275:540 *v*/*v* at a 1 mL/min flow rate with a 1 μL injection volume. The run time was 5 min and IMQ was detected using λ_exc_ = 240 nm and λ_em_ = 360 nm after 3.2 min. The calibration curve was linear over a range of 0.02–10 μg/mL (R^2^ ≥ 0.999, *p* < 0.001)

### 3.4. Construction of Pseudo-Ternary Diagram

A water titration method at ambient temperature (25 °C) was used to establish a pseudo-ternary phase diagram, enabling identifying the ME concentration range. A surfactant mixture (S_mix_) consisting of surfactant (PL) and co-surfactant (ethanol) in a 1:1 (*w*/*w*) ratio was used. The oil phase (OA) was added in the following OA/S_mix_ ratios: 0.5:9.5, 1:9, 1.5:8.5, 2:8, 3:7, 4:6, 5:5, 6:4, 7:3, 8:2, 9:1 (*w*/*w*; Table 1). The addition of water (5–95%) led to the formation of the diagram. After each addition, the mixture was gently stirred for 2 min, followed by an optical evaluation of its transparency and viscosity. Only clear, transparent mixtures with low viscosity were categorized as MEs.

### 3.5. Conductivity Studies

Conductivities of the samples prepared during the construction of the pseudo-ternary diagram were measured at a laser scattering particle size analyser (Malvern Zetasizer Nano ZS, Malvern, UK). Conductivity measurements were performed 10 times for each sample.

### 3.6. Preparation of Selected MEs with IMQ

According to the phase diagram results, six MEs (ME_0.3_, ME_0.6_, ME_1_, ME_2.5_, ME_3.75_, and ME_5.2_; Table 1) were selected and an appropriate IMQ amount (0.3, 0.6, 1.0, 2.5, 3.75 and 5.2%, respectively) was added to each one. The ME subscripts indicate their respective % IMQ concentrations.

### 3.7. ME Characterization, Rheology and Stability

Selected MEs were analysed by a laser scattering particle size analyser (Malvern Zetasizer Nano ZS, Malvern, UK) at 25 °C ± 1 °C. Measurements were carried out in triplicate at a scattering angle of 173° by using disposable polystyrene latex cuvettes. The measurements were repeated after 10 weeks to evaluate the stability of the prepared formulations [19,22,59].

Rheology experiments were conducted using a rotational rheometer (Malvern Kinexus pro) equipped with rSpace software for Kinexus. The rheometer consisted of a thermostatically controlled cone and plate tool with geometry 1/60. The apparent viscosity at a shear rate (γ) 0.01–100 s^−1^, was obtained at 25 ± 1 °C. The results were evaluated by plotting the shear stress versus the shear rate values obtained experimentally. Viscosity values at minimum shear rate are obtained from the slopes of the rheograms [28].

Samples of selected MEs were centrifuged in a MPW-260R from MPW (Warsaw, Poland) at 13,000 rpm for 30 min at 25 ± 1 °C to determine their thermodynamic stability [60]. The stability of the MEs was assessed by visual inspection of the samples stored in tightly closed tubes at room temperature in the dark for any physical changes daily in the first week and then weekly for another four weeks [61].

### 3.8. Skin Permeation and Deposition Experiments

IMQ skin permeability was evaluated using Franz diffusion cells with a permeation area of 1 cm^2^ and acceptor volume of 7.0 ± 0.2 mL. The frozen full-thickness human skin was slowly thawed at room temperature and cut into pieces. The skin fragments were mounted into the cells with the dermal side facing the acceptor compartment using Teflon holders and silicone grease. The acceptor compartment was filled with phosphate buffer saline (PBS; containing 10 mM phosphate buffer, 137 mM NaCl and 2.7 mM KCl) at pH = 7.4 with 0.0005% gentamicin sulfate to prevent microbial growth and stirred at 32 °C throughout the experiment [62]. We used PBS buffer as the acceptor because preliminary experiments and literature data suggested negligible IMQ permeation to the acceptor even if 1% albumin was added [13]. In most analysed acceptor phase samples of our preliminary experiments, the concentrations were below the lower quantification limit (0.02 μg/mL). The highest IMQ concentration that we ever found in the acceptor after 8-h application of a 25 µL dose was approximately 0.3 µg/mL (IMQ solubility in PBS was 1 µg/mL). The skin samples were left to equilibrate for 6 h, and the skin integrity was checked by measuring transepidermal water loss (TEWL, see below). Next, 25 μL of each ME was applied. Aldara cream (25 mg) was used as a positive control.

Eight hours after the application, the formulations were carefully removed from the skin (using Q-tip cotton swabs and distilled water), and the treated skin samples were left in the Franz cells for a total of 24 h to mimic the IMQ application recommended in the patient information leaflets. The acceptor phase samples were taken at 24 h and analysed for IMQ by HPLC. The tissue exposed to the donor sample was punched out, wrapped in aluminium foil, and heated to 80 °C for 1 min. The epidermis was then carefully peeled off from the dermis [62]. The epidermis, dermis, and residual skin outside the application area were weighed and extracted with 1, 2, and 3 mL, respectively, acetonitrile/ammonium acetate (5 mM, pH = 2.7) 98:2 *v*/*v* for 24 h. The extract was filtered and analysed for IMQ content by HPLC.

### 3.9. ME Effects on Transepidermal Water Loss (TEWL)

The ME effect on the skin barrier properties was evaluated by measuring TEWL using an AquaFlux AF 200 instrument (Biox Systems Ltd., London, UK). TEWL was first measured before the ME application. The upper Franz cell part was removed, and the TEWL probe was placed on the holder for 80–100 s until a stable value was obtained [53,62]. Formulations were then applied and were removed after 8 h, as described above. TEWL was recorded 1 h later (to eliminate possible adverse effects of any volatile residues on TEWL) and then for up to 25 h post-application. At all-time points, untreated skin samples were measured as negative controls [62].

### 3.10. Fourier Transform Infrared (FTIR) Spectroscopy

To check MEs’ effects on the skin barrier lipids and detect possible penetration of MEs constituents into the SC, the ME-treated SC was evaluated by FTIR. The epidermis was separated by heating the human skin at 60 °C for 2 min, and SC sheets were isolated by trypsin treatment [63]. Hydrated SC sheets, ca 1 cm × 1 cm, were treated with 40 μL of selected MEs and Aldara. Untreated SC sheets and neat MEs were used as controls. After 24 h, the residual formulation was carefully removed, the SC samples were dried with gauze and measured with a Nicolet 6700 FT-IR spectrometer (Thermo Scientific, Waltham, MA, USA) equipped with a single-reflection MIRacle attenuated total reflectance ZnSe crystal. The spectra were generated by co-addition of 256 scans collected at a resolution of 2 cm^−1^ and analysed with Bruker OPUS software [62].

### 3.11. Data Treatment

Data are presented as the means ± SD, and the number of replicates is given in the pertinent figure. One-way analysis of variance (ANOVA) with Dunnett’s post hoc test was used for statistical analysis, and *p* < 0.05 was considered significant.

## 4. Conclusions

We describe MEs based on PL and OA that improve IMQ delivery into human skin. In particular, ME_1_ with 1% IMQ delivered similar IMQ quantities to the epidermis as the commercial product with a 5-fold higher IMQ dose. IMQ was not detected in the acceptor phase after the 8 h application of ME_1_ on human skin (25 µL/cm^2^), suggesting a lower risk of systemic absorption compared to the established product. The PL components of ME_1_ were detected in the SC by FTIR, accompanied by less ordered and less tightly packed lipids. The ME_1_-induced barrier disruption recovered in less than 5 h after the formulation removal, as detected by TEWL. In conclusion, our findings demonstrate that PL/OA-based MEs could become a promising alternative for topical IMQ administration.

## Figures and Tables

**Figure 1 pharmaceuticals-15-00515-f001:**
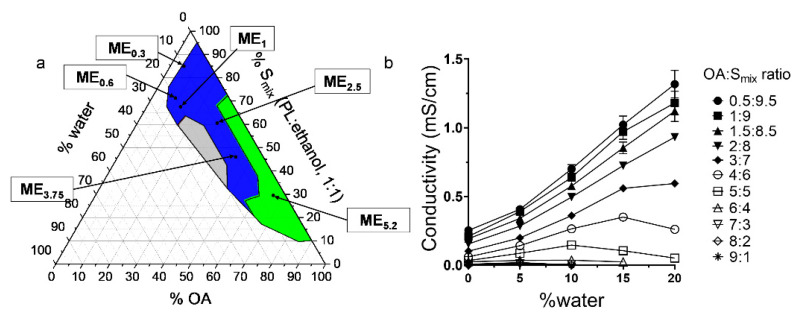
Characterization of the studied MEs. (**a**) Pseudo-ternary phase diagram; the coloured area represents the ME region (green represents *w*/*o* MEs, blue bicontinuous MEs, and grey the *o*/*w* MEs). The compositions of selected MEs are highlighted. (**b**) Electrical conductivity of MEs of varying OA: S_mix_ ratios as a function of water concentration.

**Figure 2 pharmaceuticals-15-00515-f002:**
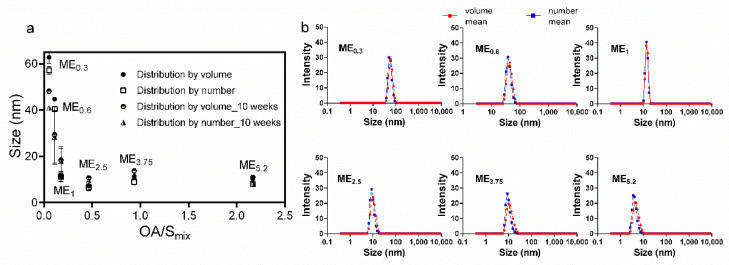
DLS characterization of MEs. (**a**) Effect of OA: S_mix_ on the size of the selected MEs immediately after preparation and 10 weeks later. Data are presented as distribution by volume and number as the means ± SD, *n* = 3. (**b**) Size distributions profiles.

**Figure 3 pharmaceuticals-15-00515-f003:**
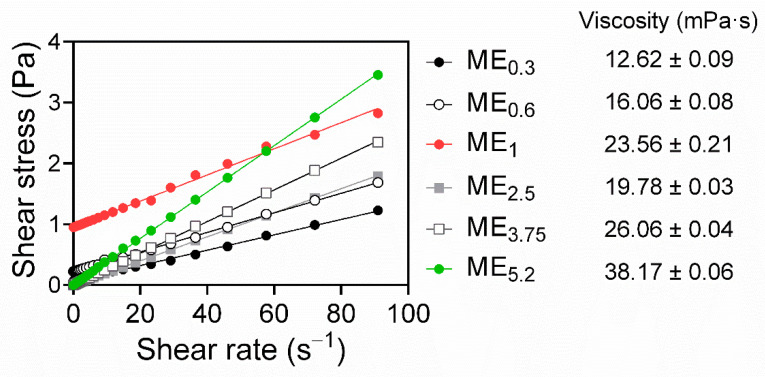
Rheograms of selected MEs, obtained by plotting shear stress versus the shear rate, show Newtonian behaviour for ME_0.3_, ME_2.5_, ME_3.75_, and ME_5.2_, and Bingham behaviour for ME_0.6_ and ME_1_ (R^2^ > 0.99). The ME viscosities are shown as the means ± SD, *n* = 3.

**Figure 4 pharmaceuticals-15-00515-f004:**
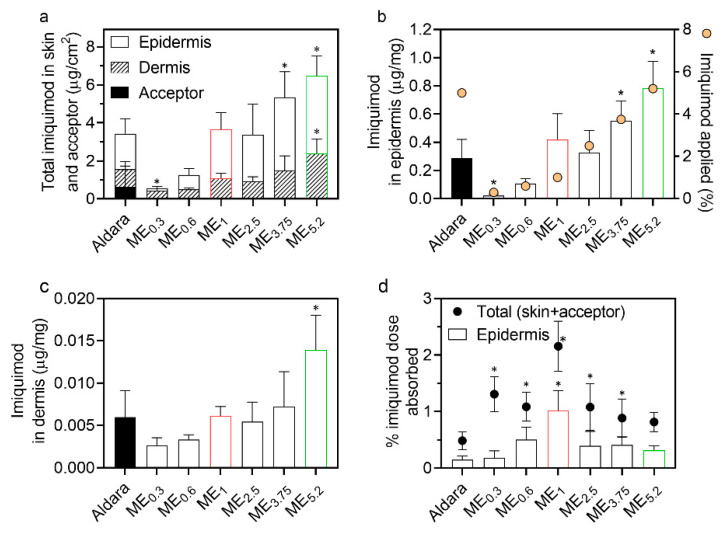
Effect of MEs on IMQ permeation and deposition in human skin in vitro. (**a**) IMQ amounts found in the epidermis, dermis, and acceptor, (**b**) IMQ concentrations in the epidermis (bars) and IMQ concentration in the formulation (circles), (**c**) IMQ concentrations in the dermis, and (**d**) proportion of the applied IMQ dose that reached the epidermis (bars) and total IMQ absorbed (circles). Data are presented as the means ± SD, *n* ≥ 4. * Statistically significant difference compared to Aldara at *p* < 0.05. Formulations selected for further experiments, ME_1_ and ME_5.2_, are highlighted in red and green, respectively.

**Figure 5 pharmaceuticals-15-00515-f005:**
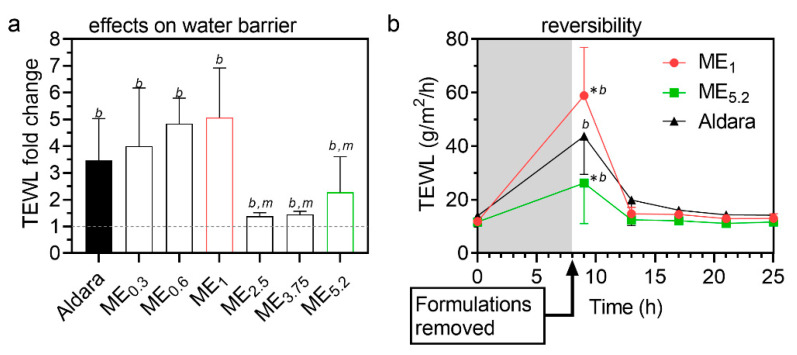
Interaction of selected MEs with the skin barrier. (**a**) TEWL fold change after/before treatment. Formulations were applied for 8 h and then carefully removed. TEWL values were obtained 1 h later to prevent interference with formulation residues. Data are presented as the means ± SD, *n* ≥ 4. *^b^* significant against baseline value (before treatment), *^m^* significant against ME_1_ at *p* < 0.05. (**b**) Rapid reversibility of TEWL values after cessation of treatment with selected formulations. Means ± SD, *n* ≥ 4. * Significant against Aldara at the same time point; *^b^* significant against baseline value (before treatment) at *p* < 0.05.

**Figure 6 pharmaceuticals-15-00515-f006:**
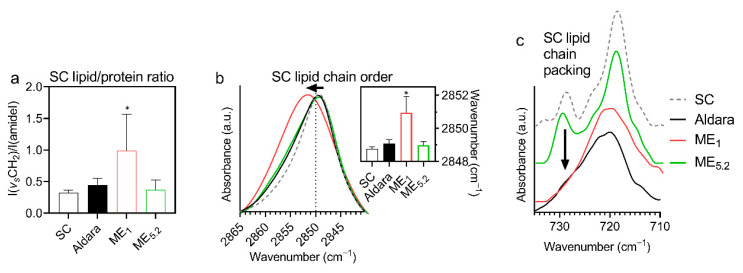
Infrared spectroscopy results of isolated human SC treated with selected formulations (means ± SD; *n* ≥ 8; * significant difference compared to untreated SC at *p* < 0.05). (**a**) Increased methylene band intensity (relative to the amide–mostly protein–bands) suggests penetration of ME_1_ components with long chains (OA and/or PL) into the SC. (**b**) A shift to high wavenumbers and broadening of the methylene symmetric stretching band indicates more disordered lipid chains in ME_1_-treated SC than in control (for example, spectra and wavenumbers in the insert). (**c**) The disappearance of the ~730 cm^−1^ component of a methylene rocking band in the SC treated with ME_1_ and Aldara indicates a loosening of the tight orthorhombic SC lipid chain packing.

**Table 1 pharmaceuticals-15-00515-t001:** Composition (weight%) of the studied MEs.

Formulation	IMQ	OA	S_mix_ (PL/Ethanol 1:1)	Water
ME_0.3_	0.30	4.50	85.20	10.00
ME_0.6_	0.60	8.00	71.50	19.90
ME_1_	1.00	11.90	67.30	19.80
ME_2.5_	2.50	27.80	59.70	10.00
ME_3.75_	3.75	41.70	44.55	10.00
ME_5.2_	5.20	61.60	28.50	4.70

## Data Availability

Data is contained within the article.

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
