# Peer review of "Phospholipid-Based Microemulsions for Cutaneous Imiquimod Delivery"

_pharmaceuticals, 2022, doi:10.3390/ph15050515_

Round 1

Reviewer 1 Report

The manuscript describes the formulation of IMQ in microemulsions made of water, oleic acid, lecithin and ethanol. Overall, this is seems to be a careful study with interesting and potentially useful results. Also the presentation is largely clear and convincing. My only suggestion is to make clearer in the section presenting the phase diagram, how exactly and with which resulution the phase ranges are established and which structures are assigned to which ranges. For example, bicontinuous ME phase has been assigned to mixtures with the following experimental parameters: (1) hydrodynamic size is ..., (2) electrical conductivity increases ... with water content, (3).....

It would be helpful to motivate these experimental readouts defining phase ranges with the putative structures of these phases: What is the hydrodynamic size of a bicontinuous phase? Why would conductivity increase with water in a w/o phase but decrease as the systrem becomes bicontinuous? Maybe cartoons would be helpful illustrating the assigments – currently, Fig. 1 is quite unclear. It also seems not optimal to mark two different ranges, of all colors, in two hues of blue.

It also seems worthwhile a comment that OA is not a simple „oil“ phase but amphiphilic itself – how does this affect the behaviour of the mixtures?

I trust that these points can be addressed well and look forward to seeing this make a nice Pharmaceutics paper.  

Author Response

Pharmaceuticals – 1687978 – Response to Reviewer 1

We sincerely thank the reviewers for their time and valuable comments, which have been used to make the manuscript stronger. Please see our point-by point responses and modifications below. We have also made appropriate changes in the revised manuscript (highlighted in yellow).

We hope you will find these modifications satisfactory and that the revised manuscript will be acceptable for publication in Pharmaceuticals.

Yours sincerely,

Georgios Paraskevopoulos

Reviewer 1

The manuscript describes the formulation of IMQ in microemulsions made of water, oleic acid, lecithin and ethanol. Overall, this is seems to be a careful study with interesting and potentially useful results. Also the presentation is largely clear and convincing.

Response:

Thank you very much for your time, and valuable comments. We have addressed all your comments and modified the unclear points. You can find our response to each of your comment below.

My only suggestion is to make clearer in the section presenting the phase diagram, how exactly and with which resulution the phase ranges are established and which structures are assigned to which ranges. For example, bicontinuous ME phase has been assigned to mixtures with the following experimental parameters: (1) hydrodynamic size is ..., (2) electrical conductivity increases ... with water content, (3).....

Response:

Thank you for this comment. We were trying different modifications to the phase diagram to make it as clear as possible (more details about the phase diagram are below). The type of each ME (o/w, bicontinuous, w/o) was addressed during the creation of the pseudo-ternary phase diagram and the conductivity measurements. Later on, only selected microemulsions of specific components were further characterized for their size, viscosity parameters and stability. To make the message clearer, an additional closing paragraph was added at the section 2.2 describing the concluding final findings and connecting them with Figure 1.

Change:

Lines 134-143: Concluding, formulations which were characterized as MEs had always less than approximately 20 % of water content regardless of the OA:Smix ratio. Furthermore, MEs were divided to o/w, bicontinuous or w/o according to their conductivity values. More specifically, MEs with low OA content and OA/Smix ratios of 0.5:9.5 – 2:8 were characterized as o/w (Figure 1a – included in grey region). MEs with medium OA/Smix ratios ranging between 3:7 – 5:5 were characterized either as o/w (when water contact was around 15%, Figure 1a – included in grey region), bicontinuous (when water content was 5-15%, Figure 1a – blue region) or w/o (when water content was < 5%, Figure 1a – included in green region). Finally, MEs with high OA/Smix ratios (6:4 – 9:1) were characterized as w/o (Figure 1a – included in green region).

It would be helpful to motivate these experimental readouts defining phase ranges with the putative structures of these phases: What is the hydrodynamic size of a bicontinuous phase?

Response:

We indeed had in mind the putative structures of these formulations and this is why we presented the DLS characterization of the prepared MEs loaded with IMQ. We know that the interpretation of the obtained data after scattering characterization of such bicontinuous systems is less straight-forward (which holds true also for situations where not well-defined, isolated droplets are observed, i.e., percolation might take place (Gradzielski, M. et al. Using Microemulsions: Formulation Based on Knowledge of Their Mesostructure. Chemical Reviews, 2021 121 (10), 5671-5740). In addition, the mechanism of particle formation in microemulsions is typically very complex and the idea that one simply transfers the size of the microemulsion, mostly the diameter of the microemulsion droplets involved, into size, shape and polydispersity of the formed nanoparticles is not accurate.

Nevertheless, the hydrodynamic size of all tested formulations (as measured in DLS characterization) is shown in Figure 2. The tested bicontinuous microemulsions had a hydrodynamic size of 57.21, 40.40, 11.08, 6.57 and 6.17nm for the ME0.3, ME0.6, ME1.0, ME2.5 and ME3.75 respectively. Obviously, the size of the bicontinuous systems varies and no direct correlation can be conducted between their hydrodynamic size and the structure.

Why would conductivity increase with water in a w/o phase but decrease as the system becomes bicontinuous?

Response:

This is a valid point concerning the structural changes happening in the microemulsions when the water content is increased. For surfactant-based aqueous microemulsions, Clausse et al. (M. Clausse, A. Zradba, L. Nicolas-Morgantini Microemulsions Systems, Dekker: New York, 1987) demonstrated that, with increasing water content, the microemulsion electrical conductivity has different changes according to four successive stages: (1) the initial nonlinear increase of conductivity reveals the existence of a percolation phenomenon that may be attributed to the inverse microdroplet aggregation; (2) the next linear increase is due to the formation of aqueous microdomains that results from the partial fusion of clustered inverse microdroplets and suggests that a W/O microemulsion is formed in the low water content region; (3) the third nonlinear curve increase indicates that the medium undergoes further structural transitions and forms a bicontinuous microstructure that is ascribed to the progressive growth and interconnection of the aqueous microdomains; and (4) the final decrease of conductivity values corresponds to the appearance of water-continuous microemulsion-type media. That is, an O/W microemulsion forms at high water content, and the progressive decrease of conductivity is ascribed to the progressive decrease of the concentration of the O/W microemulsion droplets. The manuscript was updated with appropriate comments concerning the changes observed to our systems after the addition of water.

Change:

Lines 113-116: With water addition, the conductivity of these samples increases linearly, suggesting an increasing number of water channels enabling the motion of ions and the existence of a percolation phenomenon that may be attributed to the inverse microdroplet aggregation.

Lines 130-133: The conductivity decrease at around 15% water corresponds to the appearance of water-continuous microemulsion-type media and can be explained by the formation of o/w ME with decreased concentration of the dispersed oil droplets.

Maybe cartoons would be helpful illustrating the assignments – currently, Fig. 1 is quite unclear. It also seems not optimal to mark two different ranges, of all colors, in two hues of blue.

Response:

For the Figure 1, we were trying different modifications to the phase diagram to make it as clear as possible. We have updated this figure in the revised manuscript using different colors for the different MEs as suggested and the borders between the different types bolder.

Change:

Lines 93-96: Figure 1. Characterization of the studied MEs. (a) Pseudo-ternary phase diagram; the colored area represents the ME region (green represents w/o MEs, blue bicontinuous MEs, and grey the o/w MEs). The compositions of selected MEs are highlighted. (b) Electrical conductivity of MEs of varying OA: Smix ratios as a function of water concentration.

It also seems worthwhile a comment that OA is not a simple „oil“ phase but amphiphilic itself – how does this affect the behaviour of the mixtures?

Response:

It is true that OA has an amphiphilic structure (polar carboxylic acid head and long aliphatic chain) and its role in the creation of the microemulsions can vary. Oleic acid has been used before as oil phase in different microemulsions containing IMQ (cited in our manuscript) and it is also known that, in general, fatty acids have been used as a major source for the hydrophobic part of different surfactants. Interestingly, Xu et al. reported the creation of an oleic acid containing microemulsion without the use of a classic surfactant (J. Xu, A. Jin, J. Zhao, D. Li, and W. How, Surfactant-Free Microemulsion Composed of Oleic Acid, n-Propanol, and H2O, J. Phys. Chem. B 2013, 117, 1, 450–456). Even if the authors are not emphasizing the amphiphilic character of oleic acid as the driving force of these systems, the dual role of oleic acid (as surface active molecule and oil phase) could explain their creation as n-propanol does not have a traditional amphiphilic molecular structure and shows very low surface activity for both oleic acid and water.

Change:

Lines 63-65: In addition, OA, due to its structure, has been used for the formulation of microemulsions without the use of a classical surfactant [27]

Lines 88-89: Thus, OA was chosen in this study as an oil phase with multiple action due to its amphiphilic character, ME creation ability and IMQ solubility enhancement.

Line 498: 27.      Xu, J., et al., Surfactant-free microemulsion composed of oleic acid, n-propanol, and H2O. J Phys Chem B, 2013. 117(1): p. 450-6

I trust that these points can be addressed well and look forward to seeing this make a nice Pharmaceutics paper.

Additional Note: We have corrected the following typographical error in the manuscript:

Line 112: of 0.5:9.5 – 2:8

Reviewer 2 Report

Dear Authors,

I write you in regard to the manuscript entitled Phospholipid-based microemulsions for cutaneous imiquimod delivery.

  • please, in line 141, replace "indicating" with "suggesting"
  • how was "sink condition" established?
  • title could be revised

Author Response

We sincerely thank the reviewers for their time and valuable comments, which have been used to make the manuscript stronger. Please see our point-by point responses and modifications below. We have also made appropriate changes in the revised manuscript (highlighted in yellow).

We hope you will find these modifications satisfactory and that the revised manuscript will be acceptable for publication in Pharmaceuticals.

Yours sincerely,

Georgios Paraskevopoulos

Reviewer 2

Dear Authors,

I write you in regard to the manuscript entitled Phospholipid-based microemulsions for cutaneous imiquimod delivery.

Response:

Thank you very much for your time, and valuable comments. We have addressed all your comments and modified the unclear points. You can find our response to each of your comment bellow.

please, in line 141, replace "indicating" with "suggesting"

Response:

Thank you for this suggestion. The change has been made in the revised manuscript.

Change:

Lines 155-156: The DLS characteristics of the selected MEs did not change for 10 weeks, suggesting the stability of the formulations.

how was "sink condition" established?

Response:

This is a valid point and we assured that our approach is correct in a number of preliminary experiments. We actually did a large study with IMQ-loaded liposomes that unfortunately did not meet our expectations but these experiments helped us to validate that with finite dose, PBS is an adequate acceptor. In most analyzed acceptor phase samples, the concentrations were below the lower quantification limit (0.02 μg/ml). The highest IMQ concentration that we ever found in the acceptor after 8-h application of a 25 µl dose was approximately 0.3 µg/ml, which is dissolved in PBS without any problems (the solubility was approximately 1 µg/ml, which is in agreement with Chollet, J.L.; Jozwiakowski, M.J.; Phares, K.R.; Reiter, M.J.; Roddy, P.J.; Schultz, H.J.; Ta, Q.V.; Tomai, M.A. Development of a Topically Active Imiquimod Formulation. Pharm. Dev. Technol. 19994, 35–43). We fully agree with your point that high acceptor solubility is mandatory for studying transdermal drug delivery under infinite dose conditions, but in our case (short term finite dose), PBS was adequate.

title could be revised

Response:

Thank you very much for your suggestion. We considered having a different title as you suggested but we couldn’t find something simpler and more direct than what we already have. Thus, we would appreciate if you let us to keep the title as it is.

Additional Note: We have corrected the following typographical error in the manuscript:

Line 112: of 0.5:9.5 – 2:8